# FedKLS: Federated KL-Driven Low-rank SVD Adaptation in Non-IID Data Distributions

## Abstract

Federated learning faces two key challenges: handling non-IID client distributions and reducing communication costs in adapting large models. To address these issues, we propose **FedKLS**, a framework that combines KL-divergence-based personalization with low-rank SVD-based adaptations. FedKLS chooses spectral components in a dynamical manner by mapping the heterogeneity of client distribution to the singular value spectrum, then builds specialized LoRA-style adapters, which allow aggregation at scale and client-specific specialization. Extensive experiments on 20NewsGroup and Banking77 with DistilBERT and Qwen backbones show that FedKLS achieves competitive performance compared to state-of-the-art parameter-efficient fine-tuning baselines, including LoRA, PiSSA, MiLoRA, and full fine-tuning. In highly non-IID settings ($\alpha = 0.01$), FedKLS improves F1-score by up to **11–12%** and reduces total communication cost by about **3x** over PiSSA, achieving the best trade-off between personalization and scalability. These results demonstrate the effectiveness of KL-guided spectral adaptation in federated fine-tuning of large models. Our implementation is available at: https://anonymous.4open.science/r/FedKLS.

## 1 Introduction

Transformers and large pre-trained models have revolutionized artificial intelligence (AI), yielding state-of-the-art results in natural language processing, computer vision, and beyond (Vaswani et al., 2017). However, these models often have billions of parameters, making full fine-tuning (FFT) impractical, especially in resource-constrained environments such as edge devices or distributed systems. This challenge is amplified in federated learning (FL), a mechanism that enables collaborative model training across decentralized clients while preserving data privacy (Li et al., 2019; Collins et al., 2022). In FL, clients typically have non-independent and identically distributed (non-IID) data (Ma et al., 2022), leading to critical problems that exacerbate model performance, delay convergence speed, increase communication overhead, and reduce generalization (Yuan & Li, 2022; Karimireddy et al., 2020).

Parameter-efficient fine-tuning (PEFT) has become a promising method to deal with these problems by fine-tuning only some of the parameters and freezing the rest of the pre-trained model. Low-rank Adaptation (LoRA) (Hu et al., 2022) was among the earliest approaches to deploy trainable low-rank matrices in model layers, allowing for a decrease in the number of trainable parameters with little to no performance loss. More recent work (Meng et al., 2024; Wang et al., 2024; Yun et al., 2025) has explored spectral structures using singular value decomposition (SVD), revealing that principal components tend to capture global and generalizable features, while minor components extract local or specific ones. Despite these advances, most existing PEFT methods rely on centralized training and fail to address FL-specific issues such as data heterogeneity and communication constraints.

To address these challenges, we propose **FedKLS**, a novel federated low-rank selection framework that dynamically allocates singular subspaces to clients using the Kullback-Leibler (KL) divergence between their local data and reference IID distributions. By mapping each client's distributional divergence to a subspace of the singular spectrum, IID-like clients are assigned principal components for generalization, while skewed clients adapt minor components for personalization. This design enables efficient personalization while preserving collaboration across heterogeneous clients.

Our main contributions are:

- We analyze the limitations of existing PEFT methods (LoRA, PiSSA, and MiLoRA) in federated settings and highlight how principal and minor components capture generalizable versus client-specific features under IID and non-IID distributions.

- We propose **FedKLS**, the first federated PEFT framework that explicitly links client distribution heterogeneity to spectral subspace selection, dynamically assigning singular subspaces based on the KL-divergence.

- We empirically demonstrate that FedKLS simultaneously improves communication efficiency, convergence speed, and model accuracy across diverse benchmarks, outperforming existing PEFT baselines under extremely non-IID conditions.

## 2 BACKGROUND AND RELATED WORK

### 2.1 OVERVIEW OF SINGULAR VALUE DECOMPOSITION (SVD) AND LOW-RANK MATRIX APPROXIMATIONS IN MODEL ADAPTATION

SVD is a fundamental matrix factorization technique that is widely used to decompose and compress neural network weight matrices. It decomposes a matrix $W \in \mathbb{R}^{m \times n}$ into three components: $U \in \mathbb{R}^{m \times r}$, a matrix of left singular vectors; $\Sigma \in \mathbb{R}^{r \times r}$, a diagonal matrix of singular values; and $V^T \in \mathbb{R}^{r \times n}$, a matrix of right singular vectors, where $r \ll \min(m, n)$. This decomposition,

$$W = U\Sigma V^T, \tag{1}$$

which offers a flexible way to understand the structure of learned weight matrices in large models, forms the basis for several model adaptation strategies.

In recent years, the rise of LLMs with billions of parameters has presented new challenges for efficient model adaptation and fine-tuning. Traditional full-parameter updates become computationally expensive and often impractical, motivating the development of PEFT methods. One of the most widely used PEFT methods is LoRA. Importantly, the connection between SVD and LoRA can be made explicit: if we set

$$A = U_r\sqrt{\Sigma_r}, \quad B = \sqrt{\Sigma_r}V_r^T, \tag{2}$$

then

$$AB^T = U_r\Sigma_r V_r^T, \tag{3}$$

which is exactly the truncated SVD low-rank approximation. In practice, LoRA generalizes this idea by introducing $A \in \mathbb{R}^{m \times r}$ and $B \in \mathbb{R}^{n \times r}$ as learnable low-rank matrices, instead of fixing them to singular vectors. The low-rank update is expressed as

$$W' = W + \frac{\alpha}{r}\Delta W = W + \frac{\alpha}{r}AB^T. \tag{4}$$

where $\alpha$ is a scaling factor that controls the contribution of the low-rank update. This reparameterization dramatically reduces the size of the trainable parameters from $\mathcal{O}(m \times n)$ to $\mathcal{O}(r(m + n))$, while preserving the representational feature needed for downstream tasks.

### 2.2 SVD-BASED LOW-RANK ADAPTATION METHODS

Building on LoRA's SVD-inspired foundation established above, many recent works have investigated how different spectral components can improve adaptation. PiSSA (Meng et al., 2024) initializes low-rank adapters from principal singular components to accelerate convergence, whereas MiLoRA (Wang et al., 2024) emphasizes minor components, capturing complementary subspaces that enhance generalization. Together, these methods highlight how spectral choices influence the trade-off between generalizable and specialized features. Other PEFT variants use other design objectives: DoRA (Liu et al., 2024) decomposes weights into magnitude and direction, VeRA (Kopiczko et al., 2023) uses fixed and random matrices to perform efficiently, AdaLoRA (Zhang et al., 2023) adapts rank dynamically, and QLoRA (Dettmers et al., 2023) applies quantization for memory savings. Although these methods are effective in centralized training, they are not used to deal with federated challenges such as heterogeneity or personalization.

Among the spectral-based approaches, LoRA, PiSSA, and MiLoRA are the most relevant baselines for our study, because they have SVD-based initialization strategies, which can be directly

extended to federated applications. Their theoretical basis is strengthened by the fact that SoMA (Yun et al., 2025) demonstrates that principal components represent generalizable features, whereas minor components represent domain-specific ones. These findings directly motivate our spectral allocation strategy for our FedKLS algorithm in the subsection 3.1.

# 3 METHODOLOGY

## 3.1 MOTIVATION: LoRA, PiSSA, MiLoRA, AND FedKLS

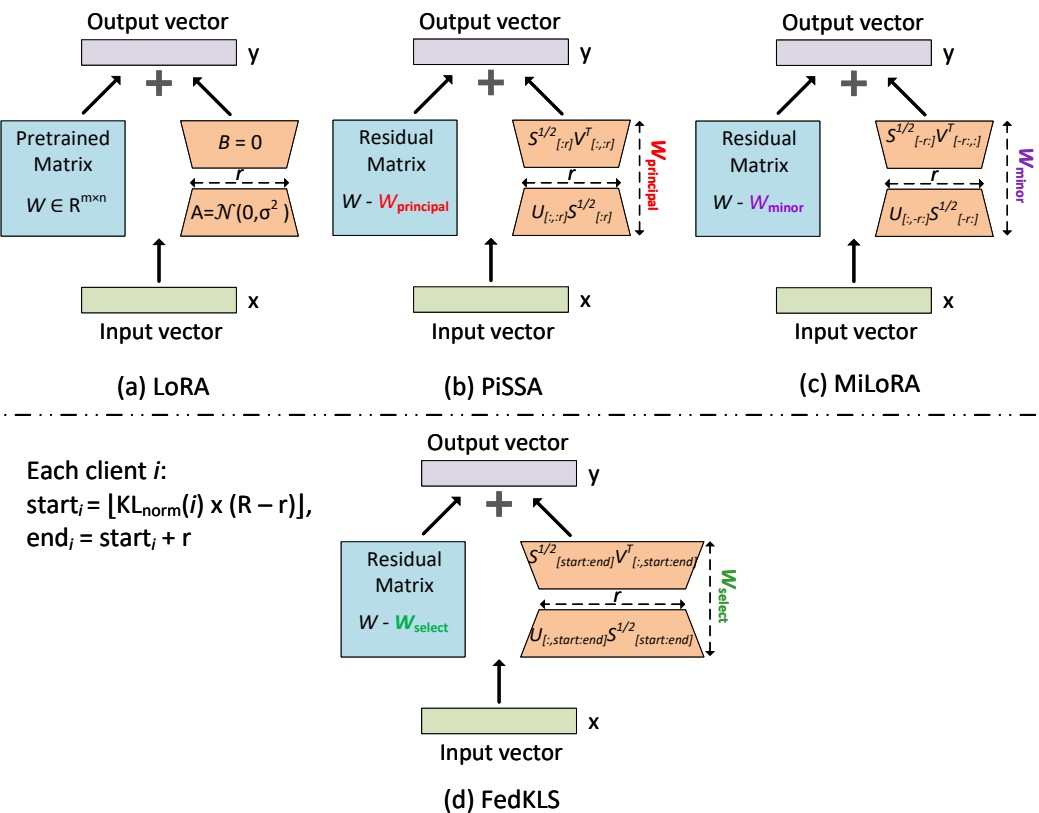

Figure 1: Comparison among LoRA, PiSSA, MiLoRA, and FedKLS. LoRA relies on random subspaces, PiSSA and MiLoRA utilize principal and minor subspaces, respectively, whereas FedKLS dynamically allocates client-specific subspaces based on the KL divergence.

Figure 1 visually contrasts the adapter initialization strategies of LoRA, PiSSA, MiLoRA, and our proposed FedKLS. **LoRA** applies random low-rank updates to the pre-trained weight matrix $W$, represented as $A \sim \mathcal{N}(0, \sigma^2)$ and $B = 0$, without leveraging any spectral decomposition. The residual matrix in LoRA is the full $W$. **PiSSA** selects the principal subspace by extracting the top-$r$ highest singular components. Here, the residual matrix $W_{residual} = W - W_{principal}$ excludes the principal components represented by $U[:, : r]\Sigma^{1/2}[: r]V^T[:, : r]$, which initialize the adapter. This leverages generalizable features for faster convergence. **MiLoRA**, in contrast, focuses on the minor subspace. The residual matrix $W_{residual} = W - W_{minor}$ excludes minor singular components $U[:, -r :]\Sigma^{1/2}[-r :]V^T[-r :, :]$, capturing context-specific information.

Rather than applying the same fixed adapters on every client, **FedKLS** dynamically determines the spectral subspace $W_{select}$ personalized to each client based on their KL divergence from the reference IID distribution. The residual $W_{residual} = W - W_{select}$ remains frozen, while the adapter adapts client-specific subspaces by selecting singular components within $[start_i : end_i]$.

## 3.2 FEDKLS ARCHITECTURE

Figure 2 illustrates the overall architecture of FedKLS, which enables personalized and adaptive low-rank model updates in FL through client-specific data distributions and SVD-based adaptation. The framework consists of three main steps:

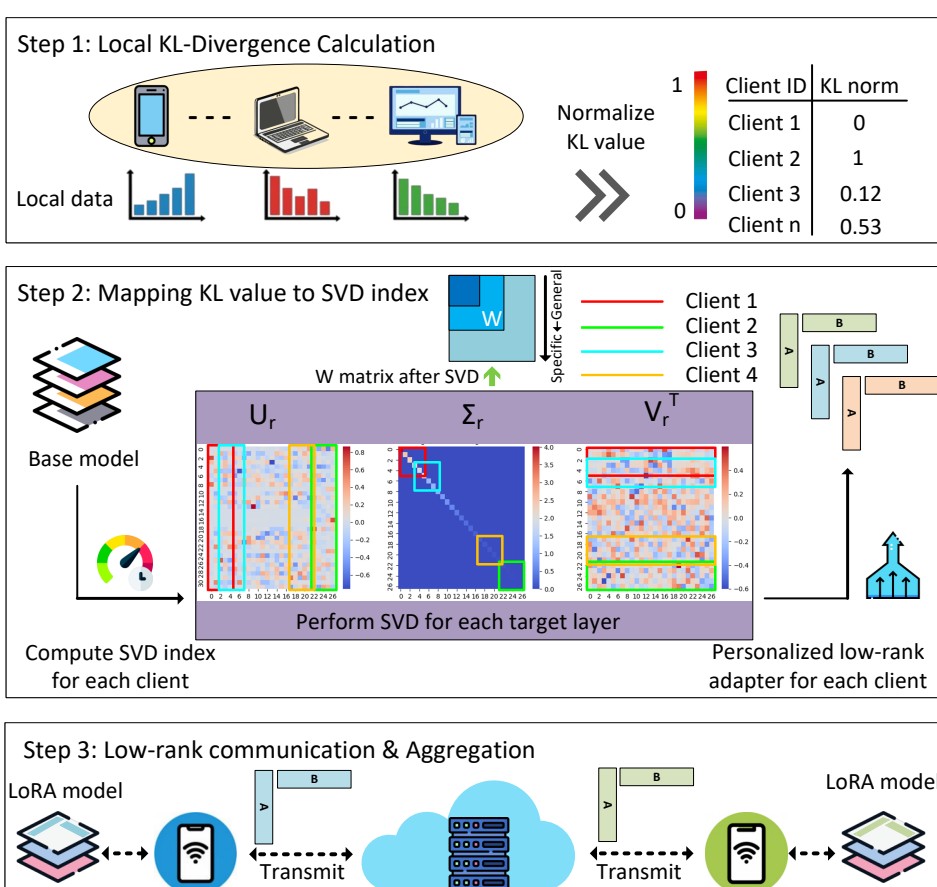

Figure 2: Overall architecture of FedKLS.

**Step 1: Local KL-Divergence Calculation.**

Each client computes the KL divergence between its local data distribution and the ideal IID distribution. These KL values are then normalized to create a client specific index that gives the level of heterogeneity between clients. The higher the KL value, the greater the divergence from the IID distribution; thus, it needs more personalized adaptation.

**Step 2: Mapping KL Value to SVD Index.** The normalized KL value is mapped to an SVD-based decomposition of the model's weight matrices. In particular, the KL value of each client is mapped to a position (SVD index) that defines the initial singular component of each linear layer in the model. According to the SVD index, a subset of singular values (from the $\Sigma$ matrix) and their vectors (from the $U$ and $V^T$ matrices) are chosen to build a client-specific low-rank adapter (A and B matrices). This mapping matches the KL divergence of each client to a suitable area of the singular spectrum: clients with smaller divergence are matched to principal components that represent generalizable features, and clients with higher divergence are matched to adapters that start with minor singular components that encode more specific, context-dependent information. For example, client 1, which has a low normalized KL value, is assigned components that are close to the starting point of the singular spectrum, thus capturing the patterns with the generalizable qualities. Client 4, on the other

hand, with a large KL divergence indicating heterogeneous data, is provided with components closer to the end of the spectrum, which are better adapted to personalization. This dynamic allocation allows each client to operate within a subspace tailored to its specific data characteristics.

**Step 3: Low-Rank Communication and Aggregation.** During communication, clients do not send the entire model parameters. Instead, only the low-rank adapter matrices $A$ and $B$ are exchanged, significantly reducing communication costs. The server aggregates these low-rank updates across clients and redistributes the global aggregated adapters. Clients then merge the aggregated adapters with their local LoRA models to continue personalized training. This process preserves communication efficiency and personalization, enabling scalable federated adaptation of large models.

Overall, FedKLS effectively balances global model aggregation and client-specific personalization by leveraging KL-divergence-guided low-rank approximations, making it well-suited for heterogeneous data environments in FL.

### 3.3 FEDKLS ALGORITHM

---

**Algorithm 1** FedKLS: Federated KL-guided Low-rank Adaptation.

---

1: **Input:** Global full-parameter model $\mathcal{M}$, rank $r$, maximum rank $R$.
2: **Output:** Low-rank models $\mathcal{M}(A, B)$ personalized for each client $i$.
   **Client side:**
3: **for** each client $i$ **in parallel do**
4:     **if** round $t = 0$ **then**                                          ▷ Initialization
5:         Compute the actual local distribution $P_i(c) = \frac{n_i(c)}{N_i}$.
6:         Compute the ideal IID distribution $Q(c) = \frac{1}{C}$.
7:         Compute KL divergence $D_{\mathrm{KL}}(P_i \parallel Q) = \sum_{c=1}^{C} P_i(c) \log(\frac{P_i(c)}{Q(c)})$.
8:         Each client $i$ sends their KL divergence $D_{\mathrm{KL}}$ to the server.
9:     **end if**
10:    **for** round $t = 1, 2, \ldots, T$ **do**                              ▷ FL Training
11:        Receive the LoRA model from the server $\mathcal{M}(A, B)$ when $t = 1$, otherwise when $t > 1$, receive the aggregated adapters $(A, B)$.
12:        **Inject adapters:** For each layer $\ell$, inject adapters $A, B$ during the forward pass.

$$W_{\ell,i}^{'} \leftarrow W_{\ell,i}^{res} + \frac{\alpha}{r} A_{\ell,i} B_{\ell,i}^{\top}, \text{ where } W_{\ell,i}^{res} \text{ is frozen.}$$

13:        **Local training:** Update adapters using stochastic gradient descent:

$$A_i^{(t+1)} = A_i^{(t)} - \eta \nabla_{A_i} \mathcal{L}_i, \quad B_i^{(t+1)} = B_i^{(t)} - \eta \nabla_{B_i} \mathcal{L}_i.$$

14:        Send updated adapters $(A_i^{(t+1)}, B_i^{(t+1)})$ back to the server.
15:    **end for**
16: **end for**
    **Server side:**
17: **if** round $t = 0$ **then**                                             ▷ Initialization
18:    **for** each client $i$ **in parallel do**
19:        Receive the KL Divergence $D_{\mathrm{KL}}(P_i \parallel Q)$ from all clients.
20:        Normalize: $KL_{\mathrm{norm}}(i) \leftarrow \frac{D_{\mathrm{KL}}(P_i \parallel Q) - D_{\mathrm{KL}}^{\min}}{D_{\mathrm{KL}}^{\max} - D_{\mathrm{KL}}^{\min}}$.
21:        Map to SVD index: $\mathrm{Index}_{\mathrm{start}}(i) \leftarrow \lfloor KL_{\mathrm{norm}}(i) \times (R - r) \rfloor$.
22:    **end for**
23: **end if**
24: **for** each communication round $t = 1, 2, \ldots, T$ **do**               ▷ FL Aggregation
25:    Server aggregates $\{(A_i, B_i)\}_{i=1}^{K}$ into $(A, B)$.
26:    Server sends aggregated $(A, B)$ back to clients.
27: **end for**

---

FedKLS enables personalized and communication-efficient FL by using KL divergence to direct client-specific low-rank adaptations of a global model. It uses the difference between the local data

Table 1: Table of notation for the FedKLS algorithm.

| Symbol | Description |
|---|---|
| **Model and Adapters** | |
| $\mathcal{M}$ | Full-parameter model. |
| $\mathcal{M}(A, B)$ | low-rank models formed by adapters $A$ and $B$. |
| $R$ | Total number of singular values or maximum rank for each layer. |
| $r$ | Fixed rank of the low-rank adapters. |
| $A_i, B_i$ | Client $i$'s low-rank adapter matrices. |
| $W_{\ell,i}^{res}$ | Frozen residual weight matrix at layer $\ell$ for client $i$. |
| $W_{\ell,i}^{'}$ | Adapted weight matrix at layer $\ell$ for client $i$. |
| $\alpha$ | Scaling factor for low-rank adapters. |
| **Data and Distribution** | |
| $C$ | Total number of classes from a dataset. |
| $n_i(c)$ | Number of samples of class $c$ in client $i$. |
| $N_i$ | Total number of samples in client $i$. |
| $P_i(c)$ | Probability distribution of class $c$ in client $i$'s local data. |
| $Q(c)$ | Reference (global or ideal IID) uniform distribution of class $c$. |
| **KL Divergence and Indices** | |
| $D_{\mathrm{KL}}(P_i \parallel Q)$ | KL divergence between client $i$'s distribution and ideal IID distribution. |
| $KL_{\mathrm{norm}}(i)$ | Normalized KL divergence for client $i$. |
| $D_{KL}^{min}$ | Minimum KL divergence among clients. |
| $D_{KL}^{max}$ | Maximum KL divergence among clients. |
| $\mathrm{Index}_{\mathrm{start}}(i)$ | Start index for chosen singular values in client $i$'s low-rank adapter. |
| **Training and Optimization** | |
| $\eta$ | Learning rate for adapter updates. |
| $\mathcal{L}_i$ | Local loss function at client $i$. |
| $\nabla_{A_i}, \nabla_{B_i}$ | Gradients of the loss function with respect to adapters $A_i$ and $B_i$. |
| $T$ | Total number of communication rounds. |

distribution of each client and the ideal IID distribution to dynamically choose singular components for building individual low-rank adapter models. This design adjusts the contribution of each client to the global model according to the heterogeneity of its data distribution.

Algorithm 1 provides the detailed procedure of FedKLS. Training and aggregation of the low-rank adapters generally follow the LoRA architecture, while the KL-guided mechanism personalizes these adapters based on client heterogeneity. At initialization, each client computes the KL divergence between its local distribution and the global reference. The server then normalizes these values into the range [0,1] and maps them to indices in the singular value spectrum, from which singular components are selected to initialize client-specific adapters. For clarity, the key notations used in Algorithm 1 are summarized in Table 1.

At initialization ($t = 0$) (Lines 4–9), clients measure how far their local distribution $P_i(c)$ deviates from the global reference $Q(c)$ by computing the KL divergence. Importantly, clients do not send their complete probability distribution to the server but only the scalar KL divergence value, thus preserving the privacy of their local data distribution while still allowing cross-client comparison. This divergence value reflects the degree of non-IIDness: clients with a small divergence are more similar to the global distribution, whereas clients with a large divergence have more specialized or skewed data.

On the server side (lines 17–23), the received KL divergence values are normalized to the range [0,1] and then mapped to indices in the singular value spectrum, which define the low-rank subspaces on which adapters are initialized. For each normalized value $KL_{\mathrm{norm}}(i)$, the server identifies a starting index and selects a corresponding block of $r$ singular components from the global model. Once these components are selected, the server applies SVD to these components to create the SVD-decomposed models $\mathcal{M}(A, B)$ and transmits them to the specific clients. This KL-guided spectral allocation achieves a sharper trade-off between global aggregation and local personalization.

Once the adapters are assigned, training proceeds in a standard federated manner (Lines 10–15), but with two key modifications. First, only the low-rank adapters $(A_i, B_i)$ are trained locally, while

the residual weights $W_\ell^{res}$ remain frozen. This greatly decreases the number of parameters being communicated and avoids overfitting to non-IID local data. Second, the personalization mechanism guarantees that every client's adapter captures a distinguishable slice of the representational capacity of the model, which is related to its distributional divergence.

During aggregation, the server collects the updated adapters and averages them to form a new set of global adapters $(A, B)$, which are then redistributed to clients (Lines 24–27). The role of aggregation here is not only to integrate knowledge across clients, but also to maintain a shared foundation that balances personalization with global generalization. Over multiple rounds, this cycle allows FedKLS to exploit both the common structure of the data and the client-specific variations, achieving a communication-efficient and distribution-aware FL process.

### 3.4 FEDKLS - CONVERGENCE ANALYSIS

We now provide a formal convergence analysis of FedKLS. The goal is to show that despite the additional personalization through KL-guided low-rank adaptation, FedKLS retains the same order of convergence guarantees as standard FL methods such as FedAvg, up to an approximation term introduced by the low-rank subspace.

**Problem setup.** FedKLS minimizes the global objective

$$F(\{\mathcal{M}_i\}) = \sum_{i=1}^{K} p_i F_i(\mathcal{M}_i), \tag{5}$$

where $K$ is the number of clients, $p_i$ are normalized aggregation weights, and each local function $F_i$ is the expected loss over client $i$'s data. Each personalized model is given by $\mathcal{M}_i = \mathcal{M}(A_i, B_i)$, where $A_i, B_i \in \mathbb{R}^{d \times r}$ are client-specific low-rank adapters.

**Assumptions.** We adopt the following standard assumptions:

- **Smoothness.** Each $F_i$ is $L$-smooth: $\|\nabla F_i(x) - \nabla F_i(y)\| \le L\|x - y\|$.

- **Bounded variance.** Stochastic gradients satisfy $\mathbb{E}\|\nabla f_i(x; \xi) - \nabla F_i(x)\|^2 \le \sigma^2$.

- **Bounded heterogeneity.** Client gradients deviate from the global gradient by at most $\|\nabla F_i(x) - \nabla F(x)\|^2 \le \zeta^2$.

- **Low-rank approximation error.** For each layer $\ell$, let $W_\ell$ be the full-rank weight and $W_\ell^{(r)}$ its best rank-$r$ approximation from SVD. Then $\Delta_{\text{SVD}} = \sum_\ell \|W_\ell - W_\ell^{(r)}\|_F^2$ denotes the cumulative error introduced by truncation. FedKLS further selects subspaces via KL divergence, so the effective error is bounded by $\Delta_{\text{SVD}} \le \gamma \sum_\ell \|W_\ell - W_\ell^{(r)}\|_F^2$, where $\gamma \in [0, 1]$ depends on the KL-based allocation.

**Convergence of FedKLS** Run FedKLS with learning rate $\eta \le 1/L$ for $T$ communication rounds. Then the average squared gradient norm of the global objective satisfies

$$\frac{1}{T} \sum_{t=1}^{T} \mathbb{E}\Big[\|\nabla F(\mathcal{M}^{(t)})\|^2\Big] \le \frac{2\big(F(\mathcal{M}^{(0)}) - F^*\big)}{\eta T} + \frac{\eta L \sigma^2}{K} + \zeta^2 + \Delta_{\text{SVD}} + \Delta_{\text{KL}}, \tag{6}$$

where $F^*$ is the optimal objective value, $\Delta_{\text{KL}}$ is a potential approximation error introduced by KL-based adaptive subspace allocation.

**Proof sketch.** The proof follows standard analysis of federated SGD. (i) $L$-smoothness ensures descent in expectation up to variance and heterogeneity terms. (ii) Averaging across $T$ rounds introduces the variation term $\frac{\eta L \sigma^2}{K}$. (iii) The heterogeneity term $\zeta^2$ captures the deviation across clients. (iv) Low-rank truncation error bounds the gradient approximation error. (v) $\Delta_{\text{KL}}$ captures errors due to imperfect spectral assignments via the KL divergence, introduced by the heuristic mapping and dynamic allocation.

# 4 EXPERIMENT

## 4.1 EXPERIMENTAL SETUP

We evaluate FedKLS on 20NewsGroup (Borkar & Dhande, 2017) and Banking77 (Srivastava, 2024), with data partitioned among clients using a Dirichlet distribution ($\alpha \in 0.1, 0.01$) (Wang et al., 2020) to simulate varying non-IID levels. Experiments are conducted with DistilBERT and Qwen1.5-0.5B, comparing against LoRA, PiSSA, MiLoRA, and full fine-tuning. As aggregation baselines, we adopt FedAvg, the standard averaging strategy in FL, and FedAWA (Shi et al., 2025), which adaptively adjusts aggregation weights using client vectors that reflect local update directions, thereby addressing data heterogeneity more effectively without requiring any proxy datasets. The main configuration uses 10 clients with a 30% participation ratio, 1000 communication rounds, 1 local epoch, and batch size 32, with scaling experiments on 20, 50, and 100 clients. Learning rates are tuned from 0.01 to 0.0001 using SGD, with adapter rank set to 32, $R = 768$ for DistilBERT and $R = 1024$ for Qwen. Training is performed on 4×A100 GPUs (Qwen, scaling) and an RTX 4090 (DistilBERT), developing based on the FedEasy framework (Kundroo et al., 2025). We report the maximum client-side F1-score (averaged across clients) over 1000 rounds, along with loss, convergence time, convergence round, and total communication cost, averaged across three runs.

## 4.2 MAIN RESULTS

Table 2: Performance Comparison on 20News and Banking77.

| Setting | | | $\alpha = 0.1$ | | $\alpha = 0.01$ | |
|---|---|---|---|---|---|---|
| Model | Strategy | Method | 20News | Banking77 | 20News | Banking77 |
| DistilBERT | | FFT | 0.794 | 0.945 | 0.810 | 0.948 |
| | FedAvg | LoRA | 0.759 | 0.931 | 0.743 | 0.896 |
| | | PiSSA | 0.788 | **0.941** | 0.802 | 0.945 |
| | | MiLoRA | 0.765 | 0.937 | 0.757 | 0.915 |
| | | FedKLS (Ours) | **0.818** | **0.941** | **0.896** | **0.963** |
| | | FFT | 0.798 | 0.945 | 0.815 | 0.946 |
| | FedAWA | LoRA | 0.758 | 0.933 | 0.752 | 0.896 |
| | | PiSSA | 0.785 | **0.943** | 0.804 | 0.941 |
| | | MiLoRA | 0.765 | 0.935 | 0.761 | 0.916 |
| | | FedKLS (Ours) | **0.820** | 0.941 | **0.894** | **0.968** |
| QWEN-0.5B | | FFT | 0.794 | 0.935 | 0.822 | 0.917 |
| | FedAvg | LoRA | 0.704 | 0.884 | 0.711 | 0.854 |
| | | PiSSA | **0.757** | **0.918** | 0.78 | 0.888 |
| | | MiLoRA | 0.721 | 0.902 | 0.751 | 0.883 |
| | | FedKLS (Ours) | 0.724 | 0.864 | **0.804** | **0.896** |
| | | FFT | 0.798 | 0.931 | 0.825 | 0.938 |
| | FedAWA | LoRA | 0.712 | 0.869 | 0.726 | 0.917 |
| | | PiSSA | **0.776** | **0.917** | 0.781 | 0.92 |
| | | MiLoRA | 0.718 | 0.886 | 0.762 | 0.905 |
| | | FedKLS (Ours) | 0.731 | 0.871 | **0.811** | **0.935** |

Table 2 compares FedKLS against LoRA, PiSSA, MiLoRA, and FFT across two datasets: 20News and Banking77. FedKLS consistently delivers the best performance across both DistilBERT and Qwen models, with particularly large gains under highly non-IID settings ($\alpha = 0.01$). For example, on 20News with DistilBERT under FedAvg, FedKLS achieves 0.896 compared to 0.743 for LoRA and 0.802 for PiSSA, while on Banking77 it reaches 0.963 compared to 0.945 (PiSSA). Even when compared with full fine-tuning (FFT), FedKLS remains competitive while requiring far fewer trainable parameters. These results confirm that KL-guided spectral selection strengthens low-rank adaptation by aligning updates with client-specific data distributions, leading to robust personalization and improved generalization.

Table 3: Communication Efficiency.

| Setting | | | Round | | Time (hrs) | | Cost (GB) | |
|---|---|---|---|---|---|---|---|---|
| Dataset | Model | Method | $\alpha = 0.1$ | $\alpha = 0.01$ | $\alpha = 0.1$ | $\alpha = 0.01$ | $\alpha = 0.1$ | $\alpha = 0.01$ |
| 20News | DistilBERT (Target F1: 0.73) | FFT | 63.33 | 126.33 | 0.282 | 0.565 | 101.79 | 203.05 |
| | | LoRA | 601 | 795 | 2.273 | 3.026 | 38.94 | 51.51 |
| | | PiSSA | 179.66 | 326.33 | 0.682 | 1.242 | 11.642 | 21.146 |
| | | MiLoRA | 440 | 601.33 | 1.666 | 2.302 | 28.512 | 38.966 |
| | | FedKLS | **165.66** | **83.33** | **0.476** | **0.246** | **10.735** | **5.4** |
| | QWEN-0.5B (Target F1: 0.7) | FFT | 31 | 41 | 1.591 | 2.159 | 345.222 | 456.59 |
| | | LoRA | 805 | 920.66 | 34.96 | 42.77 | 294.14 | 336.41 |
| | | PiSSA | **57.33** | 91.66 | **2.625** | 4.172 | **20.949** | 33.495 |
| | | MiLoRA | 387.6 | 293 | 17.52 | 13.295 | 141.65 | 107.06 |
| | | FedKLS | 339.66 | **52.66** | 12.27 | **1.931** | 124.11 | **19.24** |
| Banking77 | DistilBERT (Target F1: 0.89) | FFT | 25.8 | 68 | 0.125 | 0.752 | 41.4 | 109.29 |
| | | LoRA | 524.6 | 938.6 | 1.315 | 7.47 | 33.99 | 60.82 |
| | | PiSSA | **171.3** | 306.3 | **0.435** | 2.444 | **11.102** | 19.85 |
| | | MiLoRA | 415.3 | 674.3 | 1.049 | 5.267 | 26.913 | 43.69 |
| | | FedKLS | 284.6 | **194** | 0.669 | **0.282** | 18.44 | **12.57** |
| | QWEN-0.5B (Target F1: 0.8) | FFT | 17 | 29.66 | 0.858 | 1.091 | 189.31 | 330.37 |
| | | LoRA | 161 | 536.66 | 4.974 | 17.015 | 58.829 | 196.098 |
| | | PiSSA | **19.66** | 78 | **0.617** | 2.4 | **7.186** | 28.501 |
| | | MiLoRA | 43.33 | 234.33 | 1.317 | 7.218 | 15.834 | 85.625 |
| | | FedKLS | 51 | **39** | 1.226 | **1.238** | 18.635 | **14.25** |

Table 3 further evaluates communication efficiency in terms of convergence rounds, convergence time, and total communication cost until reach a target F1 score. The target F1 is set as the minimum score achieved by any method in each dataset–model pair, ensuring a fair comparison under consistent accuracy requirements. FedKLS significantly reduces communication overhead compared to FFT and other PEFT baselines while maintaining superior accuracy. For instance, on Banking77 with Qwen at $\alpha = 0.01$, FedKLS lowers the communication costs to 14.25 GB versus 196.098 GB for LoRA and 85.625 GB for MiLoRA. A similar trend is observed on Distilbert, where FedKLS converges in only 83.33 rounds compared to 795 rounds for LoRA. These results highlight that FedKLS not only improves accuracy but also scales efficiently in heterogeneous FL environments.

## 5 CONCLUSION

This paper proposed **FedKLS**, a federated learning framework that combines KL-divergence-driven personalization with low-rank SVD-based adaptation. By distributing client heterogeneity along the singular value spectrum, FedKLS dynamically selects spectral components and builds personalized LoRA-style adapters, enabling scalable global aggregation with efficient client-specific specialization. Experiments on 20NewsGroup and Banking77 with DistilBERT and Qwen show that FedKLS consistently outperforms state-of-the-art PEFT baselines, particularly under highly non-IID settings. These findings highlight FedKLS's effectiveness in addressing both heterogeneity and scalability challenges in FL. Future work will explore extensions to multi-modal models, handling client distribution drift, and adaptive rank selection.

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

## A RESULTS ON SCALING THE NUMBER OF CLIENTS

Table 4: Scaling clients.

| Setting | | 20 clients | | 50 clients | | 100 clients | |
|---|---|---|---|---|---|---|---|
| Strategy | Method | 20News | Banking77 | 20News | Banking77 | 20News | Banking77 |
| | FFT | 0.82 | 0.949 | 0.811 | 0.947 | 0.804 | 0.944 |
| FedAvg | LoRA | 0.701 | 0.871 | 0.694 | 0.853 | 0.595 | 0.811 |
| | PiSSA | 0.798 | 0.883 | 0.774 | 0.865 | **0.753** | 0.831 |
| | MiLoRA | 0.746 | 0.867 | 0.729 | 0.851 | 0.638 | 0.823 |
| | FedKLS (Ours) | **0.866** | **0.893** | **0.800** | **0.875** | 0.695 | **0.843** |
| | FFT | 0.819 | 0.951 | 0.809 | 0.948 | 0.805 | 0.945 |
| FedAWA | LoRA | 0.703 | 0.883 | 0.700 | 0.865 | 0.595 | 0.825 |
| | PiSSA | 0.795 | 0.895 | 0.776 | 0.877 | **0.752** | 0.838 |
| | MiLoRA | 0.746 | 0.879 | 0.731 | 0.861 | 0.634 | 0.835 |
| | FedKLS (Ours) | **0.867** | **0.905** | **0.800** | **0.887** | 0.694 | **0.855** |

Table 4 shows how model performance scales as the number of clients increases from 20 to 100 with Dirichlet $\alpha = 0.05$. As expected, all methods degrade with more clients due to rising heterogeneity, but FedKLS achieves the highest F1 score among PEFT baselines across most cases in both 20News and Banking77. PiSSA generally ranks second, while LoRA and MiLoRA drop more sharply under heavy skew. Compared to FedAvg and FedAWA with FFT, FedKLS offers the best trade-off by maintaining strong accuracy even in the 100-client case, highlighting the robustness of KL-guided subspace allocation for large-scale FL.

## B RESULTS ON LOSS PERFORMANCE

Table 5: Loss Performance on 20News and Banking77 Datasets.

| Setting | | | $\alpha = 0.1$ | | $\alpha = 0.01$ | |
|---|---|---|---|---|---|---|
| Model | Strategy | Method | 20News | Banking77 | 20News | Banking77 |
| | | FFT | 11.719 | 2.736 | 14.859 | 3.2796 |
| | FedAvg | LoRA | 12.275 | 12.067 | 15.787 | 14.534 |
| | | PiSSA | 11.839 | **10.748** | 14.44 | 11.853 |
| | | MiLoRA | 12.117 | 11.561 | 15.41 | 13.565 |
| DistilBERT | | FedKLS (Ours) | **9.362** | 11.023 | **8.728** | **10.881** |
| | | FFT | 11.689 | 2.654 | 14.887 | 3.425 |
| | FedAWA | LoRA | 12.285 | 12.037 | 15.767 | 14.527 |
| | | PiSSA | 11.852 | **10.708** | 14.363 | 11.869 |
| | | MiLoRA | 12.152 | 11.552 | 15.402 | 13.593 |
| | | FedKLS (Ours) | **9.380** | 11.044 | **8.753** | **10.867** |
| | | FFT | 11.848 | 3.407 | 14.338 | 4.214 |
| | FedAvg | LoRA | 11.769 | 5.025 | 16.31 | 6.422 |
| | | PiSSA | 11.477 | **3.88** | 15.33 | 5.65 |
| | | MiLoRA | 11.809 | 4.93 | 15.215 | 5.886 |
| QWEN-0.5B | | FedKLS (Ours) | **10.609** | 4.727 | **9.958** | **4.122** |
| | | FFT | 11.349 | 3.485 | 14.67 | 3.628 |
| | FedAWA | LoRA | 11.952 | 6.02 | 16.993 | 4.547 |
| | | PiSSA | 11.512 | **3.881** | 15.284 | 4.511 |
| | | MiLoRA | 11.715 | 5.487 | 15.83 | 5.215 |
| | | FedKLS (Ours) | **10.608** | 4.606 | **8.346** | **2.466** |

Table 5 presents the loss performance of various PEFT methods, including FedKLS, across two datasets (20News and Banking77) using DistilBERT and QWEN-0.5B backbones under different data heterogeneity levels (controlled by $\alpha$). FedKLS mostly achieves the lowest loss values, indicating stronger model fitting and adaptation compared to baseline methods such as LoRA, PiSSA, and MiLoRA.

For instance, under FedAvg with DistilBERT at $\alpha = 0.01$, FedKLS reduces the loss to 8.728 on 20News, markedly better than 15.787 for LoRA. The advantage persists across various combinations of datasets, backbones, and heterogeneity regimes, demonstrating FedKLS's robustness. These findings highlight that adaptive KL-guided spectral allocation positively impacts convergence and optimization quality, particularly under non-IID client distributions. All experiments were conducted using focal loss as the training objective to better handle class imbalance and enhance robustness in heterogeneous federated environments.

## C RESULTS ON CIFAR100

Table 6: Performance on CIFAR100 dataset.

| Setting | | | F1 score | | Round | | Time (hrs) | | Cost (GB) | |
| --- | --- | --- | --- | --- | --- | --- | --- | --- | --- | --- |
| Dataset | Model | Method | $\alpha = 0.1$ | $\alpha = 0.01$ | $\alpha = 0.1$ | $\alpha = 0.01$ | $\alpha = 0.1$ | $\alpha = 0.01$ | $\alpha = 0.1$ | $\alpha = 0.01$ |
| | | FFT | 0.591 | 0.583 | 109 | 173 | 0.46 | 0.488 | 29.386 | 46.64 |
| CIFAR100 | ResNet18 (Target F1: 0.5) | LoRA | 0.513 | 0.501 | 912 | 957 | 2.867 | 2.955 | 24.07 | 25.26 |
| | | PiSSA | 0.567 | 0.554 | 318 | 370 | 1.132 | 1.347 | 8.39 | 9.768 |
| | | MiLoRA | 0.546 | 0.542 | 566 | 582 | 1.782 | 1.999 | 14.942 | 15.364 |
| | | FedKLS | **0.57** | **0.589** | **305** | **265** | 1.032 | 0.816 | **8.052** | **6.996** |

Table 6 shows that on CIFAR100 with ResNet18, FedKLS consistently outperforms LoRA, PiSSA, and MiLoRA in both accuracy and efficiency. While full fine-tuning (FFT) achieves the highest F1 at $\alpha = 0.1$ (0.591), FedKLS obtains competitive F1 (0.57 at $\alpha = 0.1$ and 0.589 at $\alpha = 0.01$) with dramatically fewer communication rounds (305 vs. 109 for FFT, and only 265 vs. 173 at $\alpha = 0.01$), much lower training time, and the lowest communication cost ($\approx$7–8 GB versus 29–46 GB for FFT). These results highlight FedKLS's ability to maintain strong performance while significantly reducing training overhead under non-IID settings.

## D ADDITIONAL CLARIFICATIONS ON KEY DESIGN ASPECTS OF FEDKLS

### D.1 KL DIVERGENCE COMPUTATION FIXED AT INITIALIZATION

In FedKLS, the client calculates its KL divergence between the local empirical distribution and the ideal IID distribution just once, during the initialization ($t = 0$). This metric is used as a distribution-sensitive and stable proxy of client heterogeneity during training. This is assumed in most FL configurations where the distributions of data are relatively fixed. If client distributions change over time, then it would be possible to recalculate KL divergences and re-map spectral adapters every few rounds or after some specific period to maintain personalization fidelity.

### D.2 CROSS-SLICE AGGREGATION OF ADAPTER MATRICES

FedKLS aggregates the low-rank adapter matrices $A$ and $B$ across clients without explicit alignment of spectral slices since each client's adapters correspond to different singular value indices. However, meaningful aggregation is maintained for the following reasons:

- The residual weight matrices $W_{res}$ remain frozen, providing a stable base model. Thus, adapters always update relative to this fixed residual, preventing misalignment during forward passes.

- All adapter matrices share the same fixed shape and lie within a consistent low-rank parameter space. Hence, even though spectral slices differ among clients, the updates represent valid directions in the same subspace.

In summary, cross-slice aggregation takes advantage of the low-rank common parameterization to develop adapters that are compatible across clients. Rather than requiring strict alignment of singular components, FedKLS benefits from the diversity of spectral slices, enabling robust global updates while still accommodating local heterogeneity.

## E  POTENTIAL LIMITATIONS AND FUTURE ENHANCEMENTS OF FEDKLS

FedKLS is well-performing in the heterogeneous federated settings; however, there are still some limitations present. First, the KL divergence to singular subspaces mapping is a heuristic, and it adds another approximation error term ($\Delta_{KL}$), which might not allow optimal adaptation in some situations. Second, although FedKLS has better scaling compared to other PEFT baselines, there is a performance degradation with the number of clients (e.g., 100-client setup), which is indicative of challenges in very large or dynamically evolving federated configurations. Thirdly, the current analysis is limited to the visual and language tasks, which raises concerns in terms of generalizability to the multimodal or streaming contexts.

These limitations indicate some of the areas of promising future work. The exploration of adaptive strategies of subspace selection beyond the fixed KL-to-SVD mapping, such as the use of meta-learning or optimization-based allocation, with the goal of minimizing the error of the approximation of that difference, is one such direction. A different direction aims at adding strength to large-scale and dynamic participation scenarios by combining client clustering, adaptive rank assignment, or hierarchical aggregation schemes. Lastly, making FedKLS multimodal with support of multimodal backbones like vision-language models and continual FL in case of client distribution drift would make it more applicable to real-world cross-device and cross-domain applications.

