# OpenReview forum: "FedKLS: Federated KL-Driven Low-rank SVD Adaptation in Non-IID Data Distributions"
_ICLR.cc/2026/Conference — Submitted to ICLR 2026_

### Official Review · Reviewer_JQB6 · 2025-10-30

**Soundness:** 1
**Presentation:** 1
**Contribution:** 2
**Rating:** 2
**Confidence:** 4

**Summary:**

The paper proposes a new framework named FedKLS to address several challenges regarding training models in federated learning with non-IID data and high communication costs by combining KL-divergence-based personalization with low-rank SVD adaptations.

On a few benchmarks, experiments demonstrate that FedKLS significantly improves F1-scores in highly non-IID conditions and reduces communication costs compared to state-of-the-art methods.

**Strengths:**

The paper addresses an important problem of training models using highly non-iid data in Federated Learning systems.

In several results, the proposed framework outperforms a few other baselines.

Convergence analysis of the proposed method was explored, briefly.

A link to the code repo was provided. (I could not access the code, probably, due to some errors on the webpage of the repo).

**Weaknesses:**

The notation is confusing and redundant. It would be more clear to give Table 1 in the beginning of the paper, and then fix the notation used in the text and figures, accordingly.

There are a few major issues with the paper.

In half of the results, the proposed FedKLS outperforms the baseline with less cost. However, the remaining results, FedKLS underperforms the baseline with more cost. To support the superiority of FedKLS, additional analyses with various models and benchmarks should be given.

Also, the proposed convergence analysis only explores the bound on the gradient. Additional results on the convergence of optimized parameters should be given.

**Questions:**

Have you compared convergence rate of your proposed method and the other LLMs/VLMs on additional benchmarks?

In some results, FedKLS underperforms the baseline with more cost. Is this due to slow convergence or divergence of the models in those particular setups?

Could you please provide learning curves in the analyses?

---

### Official Review · Reviewer_Ln64 · 2025-10-31

**Soundness:** 3
**Presentation:** 2
**Contribution:** 2
**Rating:** 4
**Confidence:** 1

**Summary:**

The paper proposes FedKLS, a federated PEFT method that (i) computes each client’s divergence from an IID reference using KL(P||Q) over label distributions, (ii) maps the normalized KL to a contiguous slice of the global SVD spectrum per layer (principal <-> generalization, minor <-> personalization), and (iii) trains/aggregates LoRA-style adapters (A,B) while freezing residual weights. On 20News, Banking77, and CIFAR100 (ResNet18), the authors report higher F1 and lower communication vs. LoRA, PiSSA, MiLoRA, and sometimes FFT, with the largest gains at strong heterogeneity.

**Strengths:**

* **Simple, intuitive personalization signal.** Using *one scalar per client* (KL divergence) to choose a spectral subspace is a neat bridge from distribution skew → representation subspace.
* **Communication-aware design.** Communicating only ((A,B)) is standard for PEFT, and the paper focuses evaluations on rounds, time, and GB, which is appropriate in FL.
* **Targeted hypothesis.** The principal/minor spectrum ↔ generalization/personalization hypothesis is well-motivated and aligns with prior SVD-based PEFT observations.
* **Scalability experiments.** Includes client-scaling and non-IID sweeps; attempts to measure practical costs (rounds/time/GB), not just accuracy.

**Weaknesses:**

1. Aggregation correctness is under-justified.

   Clients are initialized from *different spectral slices*, yet the server averages raw $(A, B)$ across clients. The appendix asserts this is “compatible” because shapes match and $W_{\text{res}}$ is frozen, but that does not resolve potential *directional misalignment* across adapters initialized from orthogonal subspaces; simple averaging can cancel useful updates. A principled fix would align in a shared basis (e.g., project $A_i B_i^\top$ onto the SVD basis and average block-wise), or perform weight-space aggregation of $\Delta W$ after basis alignment.

2. Convergence section has a likely inequality error and hand-wavy terms.

   The paper defines $\Delta_{\text{SVD}}$ as the best rank-$r$ truncation error, then claims FedKLS “further selects subspaces” with a bound

   $$
   \Delta_{\text{SVD}} \le \gamma \sum_\ell \lVert W_\ell - W_\ell^{(r)} \rVert_F^2
   $$

   with $\gamma \in [0,1]$. This implies smaller error than the best rank-$r$ approximation, which is generally impossible. Probably the direction should be $\ge$ (or a *new* error term should be introduced). The added $\Delta_{\text{KL}}$ term is not characterized (no dependence on $r, R, K, \alpha$, etc.), making the bound non-actionable.

3. Method specification gaps.

   * Which layers receive adapters? (attn Q/K/V/O? MLP in/out? convs in ResNet?) What rank per layer, scaling $\alpha$, initialization variance, and which dtype for comms?
   * Is the SVD computed once on the (frozen) global $W$ at $t{=}0$, or per round? If $W_{\text{res}}$ is frozen forever, say so; otherwise, re-SVD should be discussed.
   * The mapping uses a contiguous slice $[{\rm start}_i:{\rm end}_i]$. Why contiguous versus, e.g., temperature-weighted sampling or mixtures? Is there enforced overlap between KL-nearby clients?

4. Label-only KL & privacy/fairness.

   * Using uniform $Q(c)=1/C$ biases the mapping when the global distribution is non-uniform; a reference aggregated (privately) or public proxy would be more faithful.
   * Even a scalar may leak info about skewed clients; privacy accounting (e.g., DP noise on the scalar) is not addressed.
   * Assigning high-KL clients to minor components could starve them of high-energy directions; fairness across clients (variance of F1) is not reported.

5. Baselines selection.

   Comparison is only to centralized PEFT baselines used in FL. The paper omits personalization baselines (Ditto, pFedMe, FedPer/FedRep, pFedHN, MOON, FedBN, FedProx/Scaffold variants with personalization heads), which would better test whether spectral routing adds value beyond standard personalized FL.

6. Metric choices and reporting.

   The text says “maximum client-side F1 (averaged across clients) over 1000 rounds”. Reporting best-over-time is optimistic; please include last-round, AUC over rounds, std over seeds, and per-client distribution (median/IQR). Communication cost (GB) must define formula (uplink+downlink? optimizer states? compression? dtype?) and participation sampling.

7. Clarity issues / small technical slips.

   In Fig. 2 Step 1, “local data is represented by $U_r$ and $\Sigma_r$” is inaccurate—those are singular vectors/values of weights, not the data. Several figure captions/text passages blur data vs. weight SVD.

**Questions:**

Check weaknesses please.

---

### Official Review · Reviewer_REwk · 2025-11-01

**Soundness:** 3
**Presentation:** 3
**Contribution:** 3
**Rating:** 4
**Confidence:** 3

**Summary:**

The paper introduces FedKLS, a novel federated learning (FL) framework that combines KL-divergence-based personalization with SVD-based low-rank adaptation. FedKLS assigns spectral subspaces to clients based on the divergence of their data distribution from a global IID reference. Clients with small KL values receive adapters for generalizability, while those with large KL values receive more personalized components. This dynamic allocation of spectral subspaces results in performance improvements compared to state-of-the-art parameter-efficient fine-tuning (PEFT) methods.

**Strengths:**

The dynamic allocation of spectral subspaces based on KL divergence and data heterogeneity is a novel contribution in the context of FL. The paper demonstrates good convergence guarantees, ensuring that FedKLS retains the same order of convergence as FedAvg. By transmitting only low-rank adapters, FedKLS significantly reduces communication costs, which is crucial for federated systems operating under bandwidth constraints.

**Weaknesses:**

1.	The fixed mapping from KL divergence to spectral subspace indices is based on a heuristic, assuming a linear relationship between KL divergence and the number of singular values. This might not always be optimal, and the method would benefit from a more flexible mapping that can adapt to different scenarios.
2.	While the aggregation of low-rank updates is feasible due to the same shape of adapter matrices, there is no guarantee that updates from distinct subspaces (e.g., principal vs. minor singular components) will aggregate effectively. This could potentially lead to issues when clients work in highly divergent parts of the singular value spectrum.
3.	The KL divergence is computed only once at the beginning of training and assumed to remain static throughout. This assumption is restrictive, as client data distributions may evolve over time (e.g., due to concept drift). Periodic recalibration of KL divergence and reassignment of spectral subspaces could be necessary for long-term performance.
4.	The computational cost of the SVD step is not discussed in detail. For large models, computing SVD for each client could become a bottleneck, particularly on edge devices with limited computational resources. A detailed breakdown of the computational cost, especially for the SVD computation, is needed.

**Questions:**

1.	Did you consider using alternative metrics of heterogeneity, such as Wasserstein distance or gradient-based divergences (e.g., Fisher divergence)? How would these affect the spectral adaptation and personalization process in FedKLS?

---

### Author Response · Authors · 2025-12-02
**Rebuttal to Reviewer Comments on FedKLS**

We thank the reviewers for the careful reading and constructive comments.

Overall, the reviews confirm that (i) the KL-guided spectral allocation idea is novel and relevant for non-IID FL, (ii) the specified hypothesis, which states that the globalization/generalization directions are reflected in the principal components, whereas the personalized/specialized ones are reflected in the minor components, is well-motivated and compatible with previous results of SVD-based PEFT, and (iii) FedKLS is performing well and saving on communication cost on a variety of challenging benchmarks, especially in extremely non-IID settings. In short, we would like to address the raised concern from each reviewer as below:

**1/ Aggregation of adapters from different subspaces.** (Reviewer REwk, Ln64)

Even though clients have varying ranges of indices, adapters are in a common global SVD basis. They just use different subspaces on the same singular vectors as though they were training different slices of the same low-rank factorization and averaging them.
Empirically, in all the experiments (Section 4.2 and Appendix), FedKLS shows convergence in the FedAvg aggregation strategy while consistently achieving higher F1 in non-IID settings.

**2/ KL is computed once at initialization.** (Reviewer REwk, Ln64)

This paper focuses on extremely non-IID yet static settings; handling data-drift or dynamic non-IID environments is outside the current scope.
However, periodically recomputing KL and reassigning slices for real-world, evolving distributions is straightforward and will be explored as future work.

**3/ SVD computational cost.** (Reviewer REwk)

We already clarified that “Once these components are selected, the server applies SVD to these components to create the SVD decomposed models M(A, B) and transmits them to the specific clients.” (Line 319-321) => **FedKLS computes the singular value decomposition only once on the server** for each selected adapter layer, since the backbone remains frozen throughout training.
**Clients never perform SVD locally**.

**4/ Alternative heterogeneity metrics.** (Reviewer REwk)

This table below summarizes the pros and cons we using KL divergence instead of other metrics:
| Metric               | What It Measures                                            | Cost                         | Client Requirements               | Pros                                                   | Cons                                                       |
|----------------------|--------------------------------------------------------------|------------------------------|-----------------------------------|--------------------------------------------------------|------------------------------------------------------------|
| **KL Divergence**    | Difference between client label distribution and reference   | Very low (O(#classes))       | Label counts or histograms        | Simple, cheap, privacy-friendly; stable scalar signal | Only label-level divergence; ignores feature geometry      |
| **Wasserstein Distance** | Earth Mover’s Distance; geometric mismatch              | High (transport problem; O(n³)) | Full samples or embeddings       | Captures geometry; meaningful for continuous dists     | Heavy computation; may violate FL privacy constraints      |
| **Fisher Divergence** | Distance via score functions (∇ log p)                     | High (score estimation)      | Access to data + gradients        | Sensitive to underlying distribution shape             | Requires ∇ log p; unstable in high dimensions              |

**5/ Privacy concerns about leaking a KL scalar value.** (Reviewer Ln64)

Only a single scalar (the KL divergence between the client’s label histogram and a fixed reference distribution) is shared with the server. This value does not expose raw samples, feature information, gradients, or model updates. It merely reflects how balanced or imbalanced the label counts are, which is significantly less sensitive than the statistics commonly exchanged in FL.

For example, in a 10-class task, many very different label histograms—such as a {30%, 10%,10%, ...} distribution and a {20%, 20%, 10%, ...} distribution—can produce nearly identical KL values when evaluated against the same reference. Thus, the mapping from label distributions to KL values is **many-to-one**, making inversion mathematically underdetermined.

**6/ Cases where FedKLS underperforms.** (α = 0.1 -> more IID) (Reviewer JQB6)

FedKLS is designed for extremely non-IID settings where clients benefit from specializing on disjoint spectral subspaces. In more IID regimes (e.g., Dirichlet α = 0.1), client distributions become similar, and the advantage of KL-guided personalization diminishes.
**This behavior is expected: when heterogeneity is low, personalization is less useful.** In these near-IID conditions, generalized approaches like FFT or PiSSA would be more helpful.

---

### Meta-Review · Area_Chair_zS6K · 2026-01-05

**Summary:**

This paper propose FedKLS, which is a new framework that combine KL-divergence and SVD to make personalized federated learning more efficient .
It choose different parts of the singular value spectrum for each client to help with data heterogeneity and reduce communication cost.

The paper is interesting, however reviewers expressed some concerns that have led to rejection of the paper. The main concerns include:

1. Some reviewers are worried because clients train on different spectral slices that are orthogonal, so averaging them at the server might cancel out the updates.
2. One reviewer found a potential problem in the convergence proof, where an inequality about truncation error seems impossible
3. One reviewer thinks it is a weakness that KL-divergence is only computed once at the start, because real data can change (this was partially addressed during the rebuttal), but for me it is also strange that one assumes a finite local dataset per client. Indeed, we make assumptions on page 7 that can be used for SGD local updates, but assume a fix data-class distribution that we can compute locally.
4. As it was noticed by a reviewer - the paper misses important personalization baselines like Ditto or FedPer, and does not give enough detail on which layers get the adapters.

**Reviewer Concerns:**

# Addressed Concerns

*  The authors partially address the concern of Reviewer REwk about static KL values.
They say that for this paper they only look at static non-IID settings, but they argue that recomputing it for "real-world, evolving distributions is straightforward"
*  Reviewer REwk was worried about SVD on edge devices. The authors clarify that SVD happens only one time on the server, not on clients, so this concern is mostly gone.

# Outstanding Concerns

*  the issue about aggregation of updates is still a problem. The authors say it is okay because it is a common basis, but they do not prove that averaging updates from completely different parts of the spectrum (principal vs. minor) actually works without cancelling each other.
*  reviewer Ln64 point out a math mistake where the error bound is better than the best possible rank-$k$ approximation. The authors did not give a clear math correction for this specific inequality in the rebuttal.
*  even though authors compare to PEFT methods, they did not add results for famous personalized FL methods like Ditto or FedPer as Reviewer Ln64 asked
*  the paper assume we can compute a fixed label distribution $P$ locally, but if we use SGD for local updates, assuming a fixed and known distribution at start is a bit strange for real FL.

**Reviewer Scores:**

Below is my estimate how the scores could have changed
*  Reviewer REwk (Initial Score: 4) - I think this reviewer might keep score at 4 or maybe move to 6.
*  Reviewer Ln64 (Initial Score: 4) - This reviewer would likely stay at 4 or go down to 3.
*  Reviewer JQB6 (Initial Score: 2) - This reviewer would likely stay at 2.

---

### Decision · Program_Chairs · 2026-01-26

Reject